# Paraneoplastic Neurological Syndromes of the Central Nervous System: Pathophysiology, Diagnosis, and Treatment

**DOI:** 10.3390/biomedicines11051406

**Published:** 2023-05-09

**Authors:** Luca Marsili, Samuel Marcucci, Joseph LaPorta, Martina Chirra, Alberto J. Espay, Carlo Colosimo

**Affiliations:** 1Gardner Family Center for Parkinson’s Disease and Movement Disorders, Department of Neurology, University of Cincinnati, Cincinnati, OH 45219, USA; luca.marsili@ucmail.uc.edu (L.M.); marcucsb@ucmail.uc.edu (S.M.); laportjh@ucmail.uc.edu (J.L.); espayaj@ucmail.uc.edu (A.J.E.); 2Department of Internal Medicine, University of Cincinnati, Cincinnati, OH 45219, USA; chirrama@ucmail.uc.edu; 3Department of Neurology, Santa Maria University Hospital, 05100 Terni, Italy

**Keywords:** paraneoplastic neurological syndromes, central nervous system, neurology, oncology, movement disorders, ataxia, epilepsy, immune-checkpoint inhibitors, CAR T-cell therapies, immune suppressants

## Abstract

Paraneoplastic neurological syndromes (PNS) include any symptomatic and non-metastatic neurological manifestations associated with a neoplasm. PNS associated with antibodies against intracellular antigens, known as “high-risk” antibodies, show frequent association with underlying cancer. PNS associated with antibodies against neural surface antigens, known as “intermediate- or low-risk” antibodies, are less frequently associated with cancer. In this narrative review, we will focus on PNS of the central nervous system (CNS). Clinicians should have a high index of suspicion with acute/subacute encephalopathies to achieve a prompt diagnosis and treatment. PNS of the CNS exhibit a range of overlapping “high-risk” clinical syndromes, including but not limited to latent and overt rapidly progressive cerebellar syndrome, opsoclonus-myoclonus-ataxia syndrome, paraneoplastic (and limbic) encephalitis/encephalomyelitis, and stiff-person spectrum disorders. Some of these phenotypes may also arise from recent anti-cancer treatments, namely immune-checkpoint inhibitors and CAR T-cell therapies, as a consequence of boosting of the immune system against cancer cells. Here, we highlight the clinical features of PNS of the CNS, their associated tumors and antibodies, and the diagnostic and therapeutic strategies. The potential and the advance of this review consists on a broad description on how the field of PNS of the CNS is constantly expanding with newly discovered antibodies and syndromes. Standardized diagnostic criteria and disease biomarkers are fundamental to quickly recognize PNS to allow prompt treatment initiation, thus improving the long-term outcome of these conditions.

## 1. Introduction

Paraneoplastic neurological syndromes (PNS) of the central nervous system (CNS) consist of neurological manifestations associated with a neoplasm unrelated to the direct invasion/metastasis in the CNS [1,2]. PNS complicates approximately 1–15% of cancers, varying with the associated cancer type [1,3,4]. PNS may precede the diagnosis of cancer by 1 to 5 years in up to 70% of patients [5,6]. PNS are thought to arise from an immune response directed against common antigens/epitopes shared by tumor cells and normal healthy cells within the CNS [7]. Conversely, in non-neurological paraneoplastic syndromes and in PNS of the peripheral nervous system, the target antigen is located outside the CNS. When PNS involve the peripheral nervous system, they cause myopathies and/or myasthenia-gravis-like syndromes. Details regarding the PNS of the peripheral nervous system have been further explored in another article in this issue. PNS of the CNS, despite being relatively rare, have been increasingly recognized and detected over time, with a constantly growing body of newly discovered antibodies and expanded interest among clinicians over the last few years [8]. Hence, PNS of the CNS are often considered in the differential diagnoses in patients presenting with acute or subacute encephalopathy once the more common causes (e.g., infections, toxic and metabolic conditions, and even functional neurologic disorders or psychiatric disorders) are excluded [9,10].

At the cellular level, PNS-associated cancers may harbor gene variants coding for onconeural proteins, particularly highly immunogenic antigens that are also expressed by CNS cells and which activate the immune system [11]. Antibodies directed against intracellular (e.g., cytoplasmic, nuclear, or synaptic) neuronal antigens are traditionally named “onconeural” antibodies and are associated with cytotoxic T cells (which are thought to exert a pathogenic role) [12]. Indeed, despite their relevant role as biomarkers, these antibodies do not have a direct pathogenic role. Conversely, antibodies against neuronal surface antigens (NSA-Abs) have a direct pathogenic role but they are less likely to be associated with cancer (they are the expression of immune system activation within the context of a systemic immune condition or disease). NSA-Abs are directed against ion channels, receptors, or other components of neural membranes [1]. The distinction between onconeural and NSA-Abs has therapeutic implications: immune treatments can be highly effective for PNS associated with NSA-Abs, while they are less effective in PNS associated with onconeural antibodies (See Figure 1).

The neural substrates of the pathophysiology of these disorders are still largely unclear [13]. While brainstem dysfunction and dysautonomia are the hallmarks of PNS, sensory neuronopathy, gastroparesis, encephalopathy, and cognitive decline may predominate [14,15]. Differential diagnoses include disorders with enhanced autonomic or cardiovascular responses, including but not limited to psychiatric, neurodegenerative, and peripheral-nervous-system-related disorders [14,15].

The traditional classification of PNS and related antibodies has been recently revised by a panel of experts which developed a new set of diagnostic criteria with the aim of improving the clinical management of these conditions [16]. Accordingly, the previously-named “onconeural antibodies” (e.g., intracellular antibodies) are now called “high-risk antibodies” (e.g., associated with cancer in >70% of cases), and the NSA-Ab are now considered “intermediate-risk antibodies” (e.g., associated with cancer in 30–70% of cases) or “low-risk antibodies” (e.g., associated with cancer in <30% of cases) (See Table 1). Moreover, PNS have been divided into different risk-related clinical phenotypes: “high-,” “intermediate-,” and “low-risk” for cancer [16]. Additionally, the panel classified different levels of evidence for PNS: definite, probable, and possible. Each individual level is obtained using a “PNS-Care Score,” namely the combination of the clinical phenotype, the antibody type (“high-”, “intermediate”, or “low-” risk), the presence of the possibly associated tumor, and the time of follow-up [16]. The panel also identified recommendations for PNS triggered by immune-checkpoint inhibitors (ICIs), a treatment class currently used to treat cancers. This classification helps clinician in having an immediate evaluation of risk of underlying oncological conditions, treatment response, and prognosis for patients with PNS.

One of the most challenging aspects of PNS of the CNS, is that they can manifest as different and frequently overlapping clinical syndromes that are often difficult to promptly diagnose or to easily categorize. The readers will find the main clinical features and related associated antibodies of the PNS of the CNS in the following subsections that will discuss the clinical conditions represented by: paraneoplastic encephalitides, rapidly progressive cerebellar syndromes, opsoclonus–myoclonus syndrome (OMS), and stiff-person spectrum disorders (SPSD) (all considered “high-risk” clinical phenotypes). There will be two more subsections describing PNS of the CNS in association with cancer treatments, such as ICIs (distinct from CAR T-cell therapy-related side effects). After having described the main pathophysiological mechanisms and clinical features of PNS of the CNS, their serological markers and tumor associations, we will focus on the corresponding diagnostic and therapeutic strategies. Our objective is to critically reappraise the current clinical features, associated neurological manifestations, and main treatments of PNS of the CNS to raise awareness among clinicians, oncologists, and general neurologists of PNS of the CNS and to provide assistance in the early diagnosis and management of these rare but life-threatening conditions. We will give particular emphasis to the most recent classification of the PNS based on the intrinsic cancer risk of the antibodies found in association with these conditions (rather than the localization of the antibody within the neural cells—e.g., neural surface vs. intracellular antibodies). We will also focus our attention on the more recent theories about the pathophysiology of the rapidly progressive cerebellar syndromes within the context of the newly-described entity of latent autoimmune cerebellar ataxia, as well as on the newly described PNS clinical entities, including but not limited to the anti-Ri/ANNA2, anti-Ma2, and anti-KLHL11 related syndromes. Finally, we will direct our attention, as never conducted before in such a review, on the extensive discussion of the PNS found in association with specific cancer treatments and their proposed management. As discussed earlier, PNS of the CNS are rare conditions but require prompt recognition and the utilization of objective diagnostic biomarkers to allow clinicians to rapidly start therapies, given that they may be treatable if diagnosed in time.

## 2. Rapidly Progressive Cerebellar Syndrome

*Rapidly progressive cerebellar syndrome*, previously known as subacute cerebellar degeneration, results from inflammation-mediated degeneration of cerebellar Purkinje cells leading to ataxia becoming severely disabling under three months [1,17,18]. Hyperacute and delayed presentations have also been described [19]. Ataxia typically manifests with gait abnormalities followed by truncal and appendicular ataxia [16]. Additional brainstem involvement includes dysarthria and oculomotor abnormalities [16,20]. Imaging is often unremarkable early in disease course as radiographic evidence of cerebellar atrophy appears in late stages [1]. 

While rapidly progressive cerebellar syndrome has been linked to “high-risk” antibodies, “intermediate- and low-risk” antibodies are being increasingly implicated. One of the best described antibodies is Anti-Yo (also known as Purkinje cell antibody (PCA)—1). Anti Yo/PCA1 is considered a “high-risk” antibody because it is highly associated (>90%) with cancer, typically ovarian or breast. It presents with a cerebellar syndrome, often preceded by a prodromal period of vertigo [21,22,23]. Tr/delta/notch-like epidermal growth factor-related receptor (DNER) is another “high-risk” antibody with >90% association with cancer, typically Hodgkin lymphoma [16,24,25]. Anti-Ri/ANNA-2 is a ‘high-risk’ antibody (>80% association with cancer, primarily breast) that is classically linked to OMS (See also Section 3), but the current literature suggests it is likely more commonly associated with rapidly progressive cerebellar syndrome [26]. Less frequently, anti-Ri/ANNA-2 antibodies may present as Bickerstaff brainstem encephalitis and with oculomotor abnormalities suggestive of progressive supranuclear palsy (PSP); however, with sudden onset or stepwise or rapid progression (to differentiate it from the classic form of PSP). [26]. More recently, KLHL11 ab has been identified as an additional “high-risk” antibody (>80% association with testicular seminoma) that causes an overlapping progressive cerebellar and brainstem syndrome, typically accompanied by sensorineural hearing loss [27,28]. This constellation of symptoms is captured with the MATCH criteria where points are allocated for male sex (1 point), ataxia (1 point), testicular cancer (2 points), other type of cancer (1 point), and hearing disturbances (1 point). This validated scoring system has high sensitivity and specificity (78% and 99%, respectively) with scores of ≥4 [29].

Beyond the above, two antibodies which have been classically associated with Lambert–Eaton myasthenic syndrome (LEMS, not a focus of this review), SOX-1 and P/Q voltage gated calcium channel (VGCC), have also been implicated in progressive cerebellar ataxia [30]. LEMS and cerebellar ataxia may co-occur in patients affected by these antibodies and the presence of ataxia portends a much higher likelihood or paraneoplastic origin than LEMS alone [31]. Finally, a few additional antibodies have been reported in paraneoplastic cerebellar syndromes. These include PCA2, mGluR1, antibodies to the intracellular 65-kD glutamic acid decarboxylase (GAD65) enzyme, anti-collapsin-response-mediated protein 5 (CRMP-5), anti-amphiphysin, anti-ANNA-3, dipeptidyl-peptidase-like protein 6 (DPPX), IgLON5, and contactin-associated protein-like 2 (CASPR2) [1,16]. It is quite possible that progressive cerebellar syndromes can be explained by the presence of these antibodies, though their descriptions are not as classic as those listed above and alternative diagnoses or false-positive results should be considered.

Recognition of alternative causes of subacute ataxia is vital as many are reversible if identified early in course. Important diseases on the differential include autoimmune processes related to thyroid disease, diabetes, gluten intolerance (celiac disease), Sjogren’s syndrome, toxic/metabolic syndromes such as vitamin deficiencies (B and E) or metronidazole toxicity, infectious processes such as varicella cerebellitis, and prionopathies [32,33,34,35,36,37]. Depending on presentation, a thorough evaluation of the aforementioned entities should be considered in patients with subacute ataxia.

Recently, Manto and colleagues have proposed the new concept of latent autoimmune cerebellar ataxia (LACA) in analogy with the latent autoimmune diabetes in adults (LADA) to underline the subtle disease course of immune-mediated ataxias, including PCD [38]. LADA is a form of type II diabetes mellitus with autoimmune features, the serum biomarker (anti-GAD antibody) is not always present or can fluctuate and tends to progress with a slow pattern [38]. The disease inevitably progresses until complete pancreatic beta-cell failure within a few years. Due to the unclear autoimmune profile, it is difficult to achieve the diagnosis in the early phase before insulin production is altered [38]. LACA has some analogies with LADA, it has a slow progression, paucity of clear-cut autoimmune features, with significant difficulty for the neurologist to achieve the diagnosis in absence of positive and significantly elevated antibody titers [38]. There are some subtle neurological features which could help clinicians towards the autoimmune and paraneoplastic nature of the cerebellar syndrome, in the early phase, before PCD overtly manifests [38]. These features, namely the cognitive fluctuation within the same day (personal observation of the authors), the presence of dizziness/vertigo, vomiting, nausea, and subtle imbalance sensations may present months before the onset of the overt PCD, and clinicians should be aware of these clinical features to warrant an early diagnosis and treatment [38].

## 3. Opsoclonus–Myoclonus Syndrome

*OMS* is a rare syndrome for which a diagnosis of definite PNS can be made without the presence of an antibody [16,39,40,41]. Clinically, OMS is described by opsoclonus (conjugate fast and multidirectional saccades without intersaccadic pauses), non-epileptic myoclonus and, variably, ataxia. With the latter, it is referred to as the triad of “opsoclonus–myoclonus–ataxia” [42]. The proposed diagnostic criteria include at least three of the following four findings: (1) opsoclonus, (2) myoclonus and/or ataxia, (3) behavioral change and/or sleep disturbance, and (4) neoplastic conditions and/or presence of antineuronal antibodies [42]. These criteria allow for flexibility in atypical presentations of OMS which can have delayed onset of opsoclonus or myoclonus and markedly asymmetric ataxia [43].

OMS is more commonly seen in the pediatric population where the syndrome is associated with neuroblastoma [44]. Neuroblastoma is detected in over 50% of pediatric OMS cases [45,46]. Despite this well described association between OMS and neuroblastoma, a specific associated antibody has yet to be elucidated. However, a neuroinflammatory process is suspected, given the cytokine and lymphocyte alterations in the cerebrospinal fluid (CSF) of these patients and the presence of CD20+ B lymphocytes and CD3+ T lymphocytes in the tumor microenvironment of OMS-associated neuroblastomas [47]. Additional suspected causes of pediatric OMS include para-infectious (e.g., varicella, influenza, human herpesvirus 6, SARS-CoV-2) inflammatory syndromes as well as familial/genetic neuro-inflammatory syndromes such as Aicardi–Goutières syndrome [43,48,49,50]. A distinguishing feature of adult-onset OMS, as compared to the pediatric population, is that it is more commonly idiopathic (~61%) than paraneoplastic (~39%) [40]. Unlike the pediatric population, in which the paraneoplastic OMS is associated mostly with neuroblastoma, the adult-onset OMS, when paraneoplastic, is associated with breast cancer, ovarian cancer, and small-cell lung cancer (SCLC). In young women, cases have also been linked to ovarian teratoma [16,42]. Additional OMS-associated antibodies reported in the literature include anti-Hu/antineuronal nuclear antigen type 1 (ANNA-1), anti Yo/PCA1, anti-Ma2, and anti-NMDAR [42]. Similar to the pediatric population, para-infectious etiologies are considered the most likely cause of non-paraneoplastic OMS.

## 4. Paraneoplastic Encephalitides

*Brainstem encephalitis* is characterized by prominent brainstem involvement followed or not by multisystem neurologic dysfunction (e.g., in association with more widespread encephalitis or rapidly progressive cerebellar degeneration, as discussed above in Section 1) [16,51]. Paraneoplastic brainstem encephalitis can present with a wide array of oculomotor abnormalities, including but not limited to vertical gaze paresis, internuclear ophthalmoplegia, nystagmus, as well as bulbar weakness and dystonias [52]. In the absence of other classical signs of PNS, it can be confused with other brainstem-localizing neurological syndromes, or with PSP-cerebellar subtype (although, in this latter case, the disease progression is slower, over years) [26,53,54,55,56].

Anti-Ri/antineuronal nuclear autoantibody type 2 (ANNA-2) and anti-Ma2 encephalitis are the “high risk” onconeural antibodies most frequently associated with brainstem encephalitis; anti-KLHL11 can be also found but less frequently [16,57]. Anti-Ri/ANNA-2 commonly presents with ataxia but also with oculomotor dysfunction including OMS and vertical gaze paresis. Abnormal movements include myoclonus in about a third of patients, parkinsonism, and cervical and jaw dystonias [26,53]. Parkinsonism with accompanying supranuclear gaze palsy and cognitive impairment has been described in several cases [26,58,59,60,61]. Taken together, Anti-Ri associated-brainstem encephalitis can mimic neurodegenerative disorders in the spectrum of parkinsonism and/or dementia with prominent brainstem and cerebellar involvement [62].

The clinical presentation of anti-Ma2 encephalitis can be highly variable. In contrast to anti-Ri/ANNA-2 antibodies, which are more common in females and highly associated with breast cancer, Ma2 reactivity is found predominantly in men and frequently linked to testicular cancer [26,59,63,64]. Along with similar oculomotor abnormalities (opsoclonus, gaze palsies), many patients develop concomitant cerebellar ataxia or diencephalic symptoms, such as excessive daytime sleepiness, cataplexy, and endocrine dysfunction [63,65,66]. More recently, cases of anti-Ma2 encephalitis have been described with a motor syndrome characterized by proximal muscle weakness, head drop, and bulbar symptoms. Rarely, patients may develop atrophy or fasciculations in the upper extremities mimicking motor neuron disease. T2/FLAIR hyperintensities may be seen on brain MRI in the corticospinal tract [63]. Concomitant Ma1 antibodies are associated with worse outcome, with more frequent brainstem involvement and ataxia; they are more common in women and in those with non-germ cell tumors [59].

*Limbic encephalitis* classically presents with the subacute onset of neuropsychiatric symptoms, including memory deficits, mood dysregulation, and behavioral changes, and is frequently associated with seizures [67]. Brain MRIs often show T2/FLAIR hyperintensity in the temporal lobes with corresponding EEG findings of localized epileptiform activity [16,59,68]. Of the “high-risk” antibodies associated with limbic encephalitis, anti-Hu/ANNA-1 is the most notable. Anti-Hu/ANNA-1 antibodies are also robustly linked to SCLC in the vast majority of cases, and the associated encephalomyelitis syndrome can cause both central and peripheral nervous system dysfunction. Of note, encephalomyelitis usually occurs with clinical impairment at various sites of the central and peripheral nervous system, also including the dorsal root ganglia, the peripheral nerve, and/or the nerve roots [16]. It is commonly associated with sensory neuropathy/neuronopathy, dysautonomia, intestinal pseudo-obstruction, as well as brainstem, cerebellar, and limbic/cortical encephalitis [69,70,71]. Documented non-SCLC malignancies have included neuroendocrine tumors, adenocarcinomas, squamous cell carcinomas, germinomas, and large-cell tumors, although these represent a minority [16,72,73]. Interestingly, a small series of eight children with anti-Hu/ANNA-1 antibodies described six cases of limbic encephalitis with negative malignancy workup, and another two cases of opsoclonus–myoclonus with underlying neuroblastoma [74].

*Anti-NMDA receptor encephalitis* is the most well-described of the autoimmune encephalitides, classically associated with teratoma (usually ovarian). Indeed, Dalmau and colleagues first described serum and CSF antibodies to the NR2B and NR2A subunits of the NMDAR in this population [75]. As observed in the larger cohort studies, the median age of onset is in the third decade of life with a strong female predominance of around eighty percent, unsurprising given the teratoma association [76,77]. Of 577 patients treated internationally who had CSF samples analyzed at the University of Pennsylvania or the University of Barcelona, 38% had an associated neoplasm (including nearly half of all females), and ovarian teratoma comprised 94% of these tumors. Extraovarian teratomas accounted for an additional 2% of the total [76].

Various malignancies have also been associated with anti-NMDAR encephalitis, most frequently in middle-aged or elderly patients, and more rarely in children. There have been reports of lung, breast, uterine, and testicular cancers as well as Merkel cell carcinoma, papillary thyroid carcinoma, renal cell carcinoma, and neuroblastoma [76,77,78,79,80]. Interestingly, Bost and colleagues were able to detect expression of the NR1 subunit of the NMDAR in five of eight tested tumor samples, including two immature teratomas, a pineal germ cell tumor with a mature teratoma component, pancreatic neuroendocrine tumor, and prostatic adenocarcinoma, suggesting an underlying mechanism for CNS autoimmunity [77]. Importantly, herpes simplex virus type 1 (HSV-1) is a known non-paraneoplastic trigger of anti-NMDAR autoimmunity, particularly after HSV encephalitis, and should be considered in the appropriate clinical context [81,82,83,84]. The underlying mechanism of this is beyond the scope of this review.

In adults, anti-NMDAR encephalitis begins with a prodrome of mood changes and positive psychotic features, such as hallucinations and delusions. In the acute phase, more severe psychiatric symptoms and memory changes become evident as well as seizures and movement disorders. Dysautonomia and central hypoventilation can be seen later in the course [1,76]. Seizures are present in about eighty percent of patients, both generalized and focal, with about half of patients developing status epilepticus. Half of these cases may be refractory or super-refractory [85,86]. The EEG in around a quarter of these patients shows extreme delta brush, characterized by diffuse, continuous rhythmic delta activity with superimposed fast activity [86]. Highly specific for anti-NMDAR encephalitis, this finding may be a poor prognostic marker, associated with prolonged hospitalization, increased disability, and higher risk of mortality. However, its prognostic value has not been consistent across all studies [87].

Movement phenomenology is frequently hyperkinetic in earlier stages, including classic orofacial and limb dyskinesias, chorea, opisthotonos. Other symptoms may follow, especially rigidity, slowness, or catatonia [1]. Children are more likely to have movement abnormalities earlier in the clinical course, which may be of unexpected phenomenology in comparison to the adult phenotype (e.g., ataxia) [76,88]. Clinical presentation does not appear to vary significantly between paraneoplastic and non-paraneoplastic anti-NMDAR encephalitis; patients with malignancy exhibit worse survival due to the cancer itself [76]. Patients without an underlying tumor to treat may be more likely to relapse [76].

In general, “intermediate-” and “low-risk” antibodies predominate among limbic encephalitides [16]. One of the most common, after anti-NMDAR, is anti-leucine-rich glioma-inactivated 1 (LGI1) encephalitis, which along with CASPR2 is part of the voltage-gated potassium channel complex (VGKCC). These antibodies may be seen alone or in combination, typically in males in the sixth or seventh decade of life [89,90,91]. Distinctive facio-brachial dystonic seizures, which are very brief in duration and occur up to 100 times per day (usually alternating from one side to the other of the body), can occur in nearly half of those with anti-LGI1 encephalitis, and usually precede cognitive symptoms, of which memory deficits are most common [92,93,94]. Seizures occur in the majority of patients with LGI1 antibodies, and radiologic evidence of mesial temporal sclerosis can be found late in the disease course [92].

Antibodies to CASPR2, and to a lesser extent, LGI1, are associated with peripheral nerve hyperexcitability (manifesting as neuropathic pain and neuromyotonia), as well as Morvan syndrome, which also features prominent neuropsychiatric symptoms, dysautonomia (especially hyperhidrosis and hemodynamic instability), and disordered sleep leading to a state once classically described as “agrypnia excitata” [89,90,95,96,97]. Those with reactivity to CASPR2 frequently present with limbic symptoms at onset, more often seizures than cognitive dysfunction, and most have limbic involvement at some point during their clinical course. Cerebellar ataxia is also common in this population [91]. Onset of symptoms can be chronic and progressive, and the presence of MRI abnormalities can be unreliable, thus patients often do not meet criteria for autoimmune limbic encephalitis [89,91,98,99]. The most common neoplastic association with VGKCC antibodies is thymoma, particularly with CASPR2 antibodies compared to LGI1, and an association with acetylcholine-receptor-antibody-positive myasthenia gravis is well-documented [89,90,100,101,102].

Other “intermediate-risk” antibodies found in limbic encephalitis are gamma-amino butyric acid receptor, type B receptor (GABA_B_R) and α-amino-3-hydroxy-5- methyl-4-isoxazolepropionic acid receptor (AMPAR).

Anti-GABA_B_R encephalitis most frequently presents as a limbic encephalitis characterized by marked seizure activity compared to other encephalitides [103,104,105]. Seizures in anti-GABA_B_R encephalitis are more likely to be tonic–clonic in comparison to other encephalitides and more likely to cause status epilepticus and refractory status epilepticus [105]. Half or more of these cases are found to have SCLC [16,103,104]. These patients tend to be older than those with non-paraneoplastic anti-GABA_B_R encephalitis, and may be more likely to present with a “classic limbic syndrome” rather than with OMS or other atypical symptoms. Anti-AMPAR encephalitis presents similarly, with most developing limbic encephalitis and some with clinical or radiological evidence of more widespread cerebral involvement [106]. Like anti-GABA_B_R encephalitis, the majority of cases are paraneoplastic, and most cases are associated with SCLC. However, anti-AMPAR encephalitis is more prevalent in female patients, and other tumor types can be observed, such as thymoma, breast cancer, and teratoma [104,107]. Of note, both GABA_B_R and AMPAR antibodies have been documented with other SCLC-associated antibodies, such as SOX-1 and amphiphysin [104,108].

## 5. Stiff-Person Spectrum Disorders

*SPSD* include stiff-person syndrome (SPS) as well as the single-limb (arm or leg) variant, termed stiff-limb syndrome (SLS), and progressive encephalomyelitis with rigidity and myoclonus (PERM). SPSD are most commonly associated with systemic autoimmunity rather than a specific malignancy. Classic SPS, characterized by progressive, lower-extremity- and truncal-predominant muscle stiffness and stimulus-sensitive muscle spasms, is most frequently associated with the “low-risk”anti-GAD65 antibodies [109,110]. GAD65 antibodies may also be associated with cerebellar ataxia, seizure, and/or limbic encephalitis; these syndromes may co-occur with SPSD [111]. Anti-GAD65 antibodies are closely linked to type 1 diabetes mellitus, autoimmune thyroid disease, celiac disease, and other autoimmune conditions, although rare malignancies have been reported in the literature [109,112]. However, other antibodies associated with SPSD can be paraneoplastic in origin, most notably amphiphysin antibodies. Underlying breast cancer is frequent in this population, which tends to be older and mostly female. Cases typically lack the lower limb predominance of classic SPS and instead have more diffuse involvement, including cervical [113]. Amphiphysin antibodies can also be associated with SCLC; in these cases, other SCLC-associated antibodies such as CRMP5, Hu/ANNA-1, or others may coexist. Interestingly, patients with SCLC may be less likely to develop SPS than those with breast cancer [114].

Patients with antibodies associated with PERM, namely glycine receptor (GlyR) and DPPX antibodies, have a lower risk for an underlying malignancy [16]. PERM varies from other forms of SPSD given the presence of myoclonus and, in many cases, hyperekplexia (exaggerated startle reaction to auditory and tactile stimuli), brainstem dysfunction, and dysautonomia, which may include thermoregulatory abnormalities and diarrhea. The latter is a feature of myoclonus or hyperekplexia associated with DPPX [115,116,117]. In severe cases, PERM may even lead to respiratory failure requiring mechanical ventilation [115,116,117]. While the majority of these cases are idiopathic, multiple cases of newly diagnosed or previously treated thymomas and B-cell lymphomas have been reported with anti-GlyR-associated PERM [115,118]. DPPX antibodies have also been linked to B-cell lymphomas [117].

## 6. Immune-Checkpoint Inhibitors, CAR T-Cell Therapies, and Related Syndromes

ICIs have completely transformed cancer treatments, thus allowing increased survival and better prognosis of numerous solid malignancies [119]. The rationale of ICIs is to boost the immune system against cancer cells (e.g., blocking the immune-checkpoint receptors PD-1 and PDL-1 on the surface of immune cells or tumor cells, respectively) (Figure 2) at the expense of immune-related adverse events (irAEs). IrAEs are generated by the inhibition of negative regulators of the immune system to primarily enhance the antitumor immunity. Hence, ICIs cause adverse effects which resemble autoimmune conditions affecting several organs and systems, including the CNS. IrAEs may involve the cardiac, integumentary, endocrine, gastrointestinal, hematological, pulmonary, renal, and musculo-skeletal systems [120]. The prevalence of neurological irAEs (n-irAEs) is highly variable and may range from 1 to 12% according to different reports; they may involve both the central and peripheral nervous systems, although the latter is more frequently implicated (central: peripheral = 1:3) [120,121,122]. The challenge for clinicians is that the syndromes associated with ICIs meet diagnostic criteria for PNS, and all alternative etiologies (e.g., carcinomatous meningitis) must be excluded [16]. Recently, consensus guidelines have been developed to appropriately classify n-irAEs [120]. Seven main syndromes have been described, four of which involve the CNS. CNS-irAEs include immune-related (ir) encephalitis, ir meningitis, ir vasculitis, and ir demyelinating diseases [120]. In some cases, n-irAEs may satisfy the clinical diagnostic criteria for PNS associated with “high-risk” antibodies [16,123]. A retrospective study has detected a significant increase of Ma2-associated PNS after the introduction of ICIs in France [124]. There has been substantial interest in discovering biomarkers of disease progression in n-irAEs. For example, patients with ir-encephalitis can show associated antiphosphodiesterase 10A-Abs [125] or an increased absolute eosinophil count [126]. However, these biomarkers and the associated autoantibodies remain of limited clinical applicability. Furthermore, a significant proportion of cases are seronegative despite extensive screening [120,123]. Accordingly, the detection of antibodies is not required for the diagnosis of irAEs. Moreover, although PNS usually precede the discovery of cancer, n-irAEs triggered by ICIs only develop when the cancer is already established and treated, in general, within a short time frame after ICIs have been started.

Therapies based on genetically modified T cells harboring chimeric antigen receptors, also known as CAR T-cell therapies, represent a powerful therapeutic strategy for several hematological cancers and have been associated with significant neurotoxicity [16]. The most aggressive and life-threatening neurotoxicity related to CAR T-cell therapies is the CAR T-cell encephalopathy [127,128]. Forty percent of patients affected by CAR T-cell encephalopathy may have severe or fatal clinical courses [129]. The pathophysiology is thought to arise from the disruption of the blood–brain barrier and the subsequent edema induced by the cytokine release stimulated by CAR T-cell therapies (Figure 3) [127,130]. Symptoms usually follow a stereotyped progression beginning with somnolence, disorientation, confusion, followed by aphasia, hallucinations, and myoclonus. Severe cases progress to generalized seizures and encephalopathy possibly leading to coma and death if not promptly recognized and treated. While not considered part of the PNS, CAR T-cell encephalopathy should be distinguished from other paraneoplastic encephalitides (see Section 4).

## 7. Diagnosis

The clinical diagnosis of PNS requires the exclusion of other, more frequent causes, namely infectious and non-neoplastic-induced autoimmune disorders, cancers (including focal lesions as well as carcinomatous meningitis), rapidly progressive neurodegenerative disorders (e.g., prion disease, dementias, and motor neuron diseases which may present similarly to anti-Ma2-associated syndromes), and toxic/metabolic conditions [16]. In general, a neurological syndrome with a subacute onset should raise the index of suspicion for PNS. The presence of encephalopathic features associated with cognitive fluctuations (even within the same day) is another clue suggesting a possible autoimmune/paraneoplastic process. As discussed above, the likelihood for underlying cancer is higher in the context of “high-risk “antibodies compared to “intermediate-” and “low-risk” antibodies [1,122] (Figure 4). In some cases, specific clinical phenotypes and related isolated antibodies may also suggest the possible association with a given cancer, as, for example, in limbic encephalitis [16]. Intracellular antigens originating from tumor cells may trigger the immune response, and, consequently, the isolated “high-risk” antibodies usually target intracellular antigens found in the nucleus or cytoplasm or even in the intracellular side of the synaptic membrane [122]. The diagnostic certainty of PNS ranges from possible, to probable, to definite, according to the 10-point “PNS-Care Score” (0 = lower/absent probability; 10 = greatest probability) [16]. Definite PNS has the highest score (≥8) and is characterized by a “high-risk” clinical phenotype and antibody, as well as confirmation of a specific cancer [16]. The only exception is represented by OMS associated with neuroblastoma/small cell lung cancer where no specific antibodies are recognized [16,39]. A recent retrospective, multicenter study showed how misdiagnosis of autoimmune encephalitis may be frequent, even at specialized centers [10]. Red flags suggesting possible alternative diagnoses are a chronic and insidious disease progression, a non-specific or false positive serum antibody titer (not tested or confirmed on CSF), and non-compliance with the diagnostic criteria [10]. Other diagnoses that may mimic PNS are functional neurologic disorders (25%), neurodegenerative disorders (20%), and psychiatric diseases (18%), followed by brain neoplasms (10%), and other causes (17%) [10].

The laboratory diagnosis should start with the search for common antibodies. If absent, specialized tissue- or cell-based assays may be employed in specific second-level laboratories to screen for less common antibodies [131,132,133,134]. The widely used commercial kits for antibody detection during screening may be associated with false-positive or -negative results (in particular, for kits assessing CV2/CRMP5, Ma2, Yo, and SOX1 antibodies) [16]. The authors of this review use the kit and evaluation algorithm from the Mayo Clinic Laboratories [135], based on the execution of enzyme-linked immunoassays, radioimmunoassays, or immunofluorescence assays; if they yield a positive result for a given antibody, the immunoblotting for that antibody is performed to confirm its presence. Additionally, the diagnostic criteria endorse the usage of two different techniques to confirm the results [16]. Seronegative results and false positive results are common and are related to the techniques as well as to the state of development of the field [131,136]. A false positive finding may be suspected by the presence of an atypical clinical presentation, or when the antibodies are isolated in serum but not in CSF, or when the titers are very low [122,131]. Hence, when in doubt, clinicians should always test CSF. Antibodies to LGI-1 may be an exception as they are often absent or present only in low-titers in CSF, and more frequently isolated in serum [137]. The diagnosis of n-irAEs of ICIs can be especially difficult, generally requiring a clinical presentation mimicking PNS, with the exclusion of other more frequent causes (e.g., cancer metastasis, infectious diseases, and radiation therapy or chemotherapy side effects) as well as the evidence of CNS inflammation (e.g., imaging/CSF/neurophysiological studies associated with improvement after immune-treatments, and/or with ICIs discontinuation, or as demonstrated on biopsy) [120,138].

Importantly, nearly 80% of PNS patients show positive diagnostic screening for tumors at the initial assessment [1]. These tumors might be identified with imaging including CT, duplex ultrasound, FDG-PET, and MRI [1,139]. In selected conditions (for example germ-cell testicular neoplasms), imaging may be negative, and tumors are only revealed by histological examination [140].

## 8. Treatment

In PNS it is important to distinguish between the treatment of the underlying tumor, the treatment of the tumor-induced immune response, and the symptomatic treatment of the various PNS-related symptoms. Although not strictly within the aim of this review, the first step of PNS treatment is the oncological treatment (systemic or surgical) of the underlying tumor, when diagnosed, followed by the administration of immune treatments, when required. For example, patients with testicular germ-cell tumors and anti-Ma2 encephalitis, may benefit from radical orchiectomy followed by steroid therapy, with about 35% of cases showing good treatment response [141]. Paraneoplastic choreas, on the other hand, tend to have a worse prognosis (except for those associated with LGI1 and CASPR2 antibodies) [142]. In cases of negative work-up for malignancy, intensified tumor screening is warranted. Usually, if primary screening is negative, it is recommended to repeat tumor screening after every 3–6 months and then subsequently every 6 months for up to 4 years [143]. 

Regarding immune therapies, the first-line drugs currently used to suppress the immune system are intravenous (IV) steroids, IV immunoglobulins (IVIg), and plasma-exchange [1]. If first line interventions fail, second-line options include rituximab (anti-CD20 receptor monoclonal antibody, expressed on B cells), cyclophosphamide (DNA alkylating agent), and other compounds as mycophenolate mofetil and azathioprine [122]. In all cases, intensive oncological follow-up together with a strict neurological evaluation are required, and a multidisciplinary team including oncologists and neurologists is indispensable [144]. To date, high-level evidence is scarce on how to manage PNS. Management guidelines come from single-center studies, case series, and expert opinion. To the best of our knowledge, only two randomized clinical trials have been conducted on the efficacy of IVIg in stiff-person syndrome [145] and in patients with LGI1/CASPR2-Ab-associated epilepsy/encephalopathy [146]. Although they had a low sample size, these two trials showed markedly positive results, supporting the use of IVIg as first-line treatment in these conditions. The management of patients with n-irAEs should be conducted according to the National Comprehensive Cancer Network guidelines [147]. These guidelines suggest holding ICIs and starting IV steroids, which may be followed by IV immunoglobulins and/or plasmapheresis, if needed. Other treatments (including second-line immunosuppression) may be considered in selected cases. However, many uncertainties remain, including whether to restart ICIs after clinical improvement and the extent of treatment of the underlying oncological disease [148].

Symptomatic treatment should be considered and may vary based on the underlying neurological symptoms associated with PNS. Patients may benefit from antiseizure medications and antipsychotics for the management of hallucinations, delusions, and other psychotic features if present [10]. Myoclonus can be treated with piracetam or levetiracetam, and rigidity can be treated with muscle relaxants, such as benzodiazepines or baclofen [1,122]. Other associated symptoms may be treated with specific symptomatic drugs, namely botulinum neurotoxin injections for dystonia [1], levodopa for parkinsonism [149], and dopamine-depleting agents for chorea [142].

## 9. Discussion

Paraneoplastic neurological syndromes pose a unique challenge to the neurologist. Our understanding of the phenomena, which lie at the intersection of neuroscience, immunology, and cancer biology, is constantly evolving. Although rare, with an observed incidence of approximately 0.2 to 1 per 100,000 persons per year, PNS is being diagnosed more frequently [11,16,150]. The neurologist plays a critical role in this process: rapid identification of PNS can lead to earlier diagnosis of the underlying malignancy. Likewise, knowing the most common paraneoplastic syndromes can be useful to identify false positive antibody testing, avoid misdiagnosis, and shift focus to alternative diagnoses, such as adverse reaction to medications, CNS metastasis, and others [137,151,152].

The underlying mechanism of PNS appears to be expression of the autoantigen by the associated neoplasm, ultimately triggering a CD8+ cytotoxic T-cell-mediated response against intracellular antigens or resulting in autoantibodies binding directly to neuronal surface antigens. Symptomatology generally corresponds to the affected brain region (e.g., Purkinje cell antibodies causing rapidly progressive cerebellar syndrome) or receptor type (e.g. GABA_B_R antibodies causing intractable epilepsy) [21,22,23,103]. Therefore, clinical manifestations of a PNS alone can often (but not always) suggest a specific antibody and/or neoplasm, or a short list of differential diagnoses. Among well-known examples of this are OMS and pediatric neuroblastoma (although no associated antibody is known yet) [44,45,46] as well as encephalomyelitis, sensory neuronopathy, and intestinal pseudo-obstruction, suggestive of SCLC and anti-Hu/ANNA-1 antibodies [70].

With this paradigm, we have discussed multiple overlapping clinical phenotypes usually associated with neuronal autoantibodies, along with their risk of associated neoplasm according to the most recently updated diagnostic criteria [16]. Among the “high-risk” phenotypes outlined in this review are rapidly progressive cerebellar syndrome, OMS, and some forms of encephalitis. The former is typically characterized by truncal ataxia followed by appendicular ataxia, and frequently brainstem symptoms as well. A subacute, progressive ataxic syndrome in the proper clinical context should therefore raise suspicion for a PNS related to Hodgkin lymphoma (anti-DNER), breast cancer (anti-Ri/ANNA-2), testicular cancer (anti-KLHL11), or others [24,26,27]. Encephalitis, which may variably present with memory deficits, psychosis, and seizures, can be associated with antibodies to Hu/ANNA-1 or Ma2, suggesting underlying SCLC or testicular cancer, respectively, or can alternatively be associated with “intermediate-“risk (anti-GABA_B_R, AMPAR) or “lower-risk” antibodies (anti-LGI1, CASPR2) [16]. OMS, in addition to neuroblastoma, can be linked to breast and ovarian cancer, as well as SCLC, and can be associated with various “high-risk” autoantibodies (Ri/ANNA-2, Hu/ANNA-1, Ma2, Yo/PCA1, etc.) [42]. “Intermediate-” and “low-risk” syndromes, such as encephalitis and SPSD, can also be paraneoplastic, the former when associated with anti-Ri/ANNA-2 (breast cancer) or anti-Ma2 (testicular cancer) antibodies, and the latter when associated with anti-amphiphysin (SCLC) antibodies [26,59,63,114]. ICIs can also induce encephalitis and can be associated with “high-risk” antibodies. CAR T-cell therapy neurotoxicity, though not truly a paraneoplastic phenomenon, is an important diagnostic consideration in the encephalopathic cancer patient, and this can be fatal if unrecognized [127,128,129].

When PNS is suspected, a thorough evaluation is warranted. A careful history should ensure an appropriate time course for development of symptoms as well as evaluate for clues to an alternative diagnosis. Laboratory workup, including basic CSF studies (e.g., cell count, proteins, etc.), IgG index, and oligoclonal bands, can provide evidence of an inflammatory etiology. Antibody testing in both CSF and serum testing is recommended to increase sensitivity as well as to avoid false negatives (although some antibodies may be more likely to appear in one or the other) [16]. MRI may show T2/FLAIR hyperintensity in the temporal lobes in limbic encephalitis or may show evidence of multifocal encephalomyelitis or, alternatively, may find evidence of metastatic disease or other etiology for symptoms [98]. Electroencephalography can also strengthen the case for a paraneoplastic or idiopathic autoimmune etiology in the absence of a clear structural abnormality, for example, temporal slowing or epileptiform activity in the case of limbic encephalitis [98]. Appropriate malignancy workup can be guided by the presenting symptoms and antibody profile discovered and frequently include a whole-body CT and/or a PET scan or other dedicated imaging studies, as appropriate. Clinicians may find the antibody prevalence in epilepsy and encephalopathy (APE2) score useful in integrating clinical, imaging, neurophysiological, and laboratory data in the assessment of autoimmune/paraneoplastic encephalitides [136]. The PNS Care score can be used to classify possible, probably, and definite PNS [16].

Once diagnosed, the primary management for PNS is treatment of the underlying cancer. However, first line acute immunotherapy is frequently steroids, IVIg, or plasma exchange. Second-line therapies include B-cell depletion (generally rituximab) and cyclophosphamide [1,122]. In seronegative cases meeting criteria for PNS, empiric treatment may be considered given the clinical implications. Clinicians should be aware of these rare but disabling and potentially fatal conditions, which often become part of the differential diagnosis of acute/subacute encephalopathy once the more common infective and toxic-metabolic causes have been ruled out.

Previous research work in the field of PNS has led to great strides in our understanding of a complex entity that was unknown until relatively recently. However, there is still much exploratory work to be done. Many of the seminal clinical studies within the field of neural immunology and PNS were performed ten or more years ago [4,21,59,69,70,75,103,104]. While these studies were high in quality, they largely preceded the widespread availability of commercial antibody testing that is currently available and perhaps much of the awareness of these conditions among clinicians who are not subspecialists in neurology, neuro-oncology, or neuroimmunology. It is therefore plausible that, as more patients are evaluated for PNS, our understanding of their manifestations and the diversity of their phenotypes will change. To date, there is minimal data on the treatment of PNS, owing largely to the relative rarity of PNS generally, and especially of individual syndromes. The largest series that have been published have had only tens to hundreds of cases, even at large referral centers. Additionally, useful study endpoints would be difficult to define given the variability in pathology, location, staging, and prognosis of associated cancers, particularly when the cancer is life-limiting or requires treatment with prominent adverse effects.

Multiple questions remain for PNS. First, the syndromes remain difficult to diagnose even in specialized centers, which require invasive, extensive, and costly examinations, including but not limited to CSF analysis and neuroimaging studies, as discussed above. Second, they frequently pose a clinical dilemma for clinicians in cases of low-positivity or absent antibodies, raising questions as to whether empiric treatment is worth the risk in the face of diagnostic uncertainty [9,10]. Third, these conditions have introduced a new era in the field of neurology represented by the immune landscape of neurology: the same biomarkers (antibodies) may be associated with multiple conditions, some originally interpreted as neurodegenerative and/or untreatable. This is the case for CASPR2 and IgLON5-related syndromes, which can present even over multiple years and without evidence of relevant clinical or neuroimaging findings, other than the autoantibody itself [10,89,153,154,155,156,157,158]. Currently, researchers are investigating if certain antibodies may be the cause or the consequence of neurodegenerative diseases [159,160,161]. This specific topic is discussed in another manuscript in the present issue [162]. However, these observations may often lack of CSF confirmation, and have raised significant debate in the scientific community [137]. We are still in the initial phase of this new era and many antibodies and related pathogenic mechanisms are yet to be uncovered. Additionally, the deeper knowledge of immunology and its associated pathogenic mechanisms will completely change the way we currently imagine and categorize diseases within the field of neurology, thus allowing a paradigm shift from the old clinicopathology-based nosology of neurodegenerative disorders (e.g., Alzheimer’s and Parkinson’s diseases, in which the gold standard is the autoptic confirmation) and finally focus on their underlying biological mechanisms (e.g., genetics, immunology, proteomic, metabolomic, etc.) [163].

With the present review, we have provided an updated summary of PNS of the CNS according to the most recent clinical and biomarker-based diagnostic criteria. This includes the new category of ICI-induced n-irAEs as well as an overview of the neurologic complications of CAR T-cell therapies. We have described the diagnostic and therapeutic work-up when investigating with probable, possible, and definite PNS based on clinical findings, neuroimaging and antibody testing, and we have discussed the role of specific antibodies in confirming the diagnosis of PNS and guiding the search for a hidden cancer. We finally summarized first- and second-line immune-treatments as well as the symptomatic treatments to relief patients’ symptoms. In seronegative cases meeting criteria for PNS, empiric treatment may be considered given the clinical implications. Clinicians should be aware of these rare but disabling and potentially fatal conditions that often enter in the differential diagnosis of acute/subacute encephalopathy once the more common infective and toxic-metabolic causes have been ruled out.

The present review has some limitations. In particular, due to its narrative nature, it lacks a systematic and statistical approach to examine the existing literature on the topic. Additionally, it does not address paraneoplastic syndromes of the peripheral nervous system or other autoimmune neurological syndromes covered in other manuscripts belonging to this issue. Future studies should investigate the important topic of autoimmune neurology in an omnicomprehensive and systematic fashion.

## 10. Conclusions and Next Steps

For the near future, we envision a more standardized dissemination of validated kits for the detection of antibodies, the discovery of new reliable biomarkers for disease progression, and biomarkers for prediction of treatment response for ICI-induced n-irAEs and CAR T-cell therapies. It is our hope that the global scientific community will invest in the conduct of multicenter clinical trials to test better treatments for each one of these conditions. It is important to continue working on creating more standardized clinical diagnostic criteria and on the identification of a universal biomarker able to quickly and easily recognize the presence of PNS. Prompt recognition and treatment initiation stands to make a critical difference in the long-term outcome of these disabling conditions.

## Figures and Tables

**Figure 1 biomedicines-11-01406-f001:**
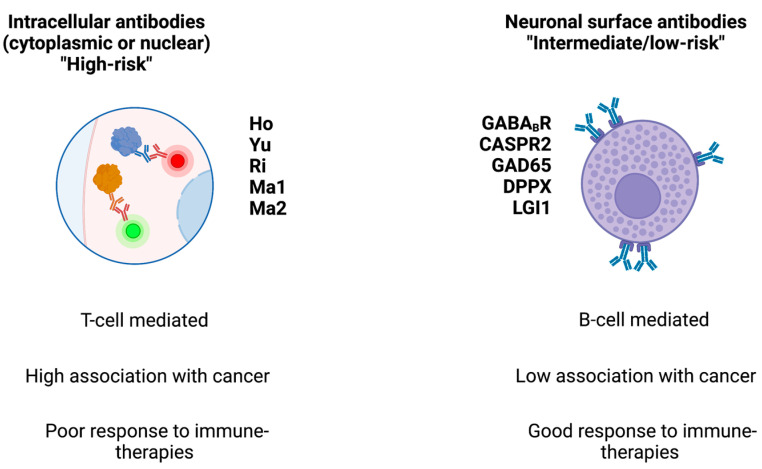
Main classes of antibodies, their associated cancer risk, and response to therapies. The figure schematizes the two main classes of antibodies found in paraneoplastic syndromes.

**Figure 2 biomedicines-11-01406-f002:**
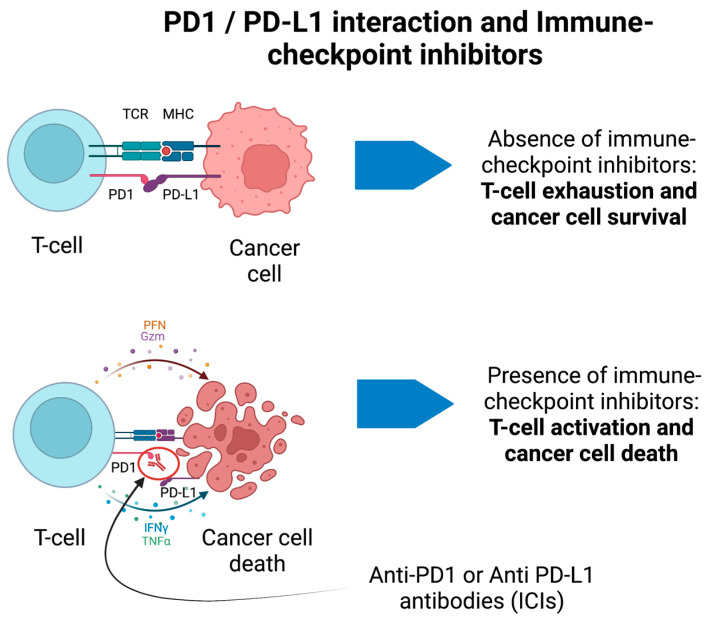
PD1/PD-L1 interaction and immune-checkpoint inhibitors for cancer treatment. In the absence of immune-checkpoint inhibitors, the binding between PD1 and PD-L1 on T-cells and cancer cells, respectively, prevents the activation of T-cells. In presence of anti-PD1 antibodies (ICIs), the T-cell is activated against cancer cells and promotes their death through different immune-mediated pathways. These may generate immune-mediated adverse events.

**Figure 3 biomedicines-11-01406-f003:**
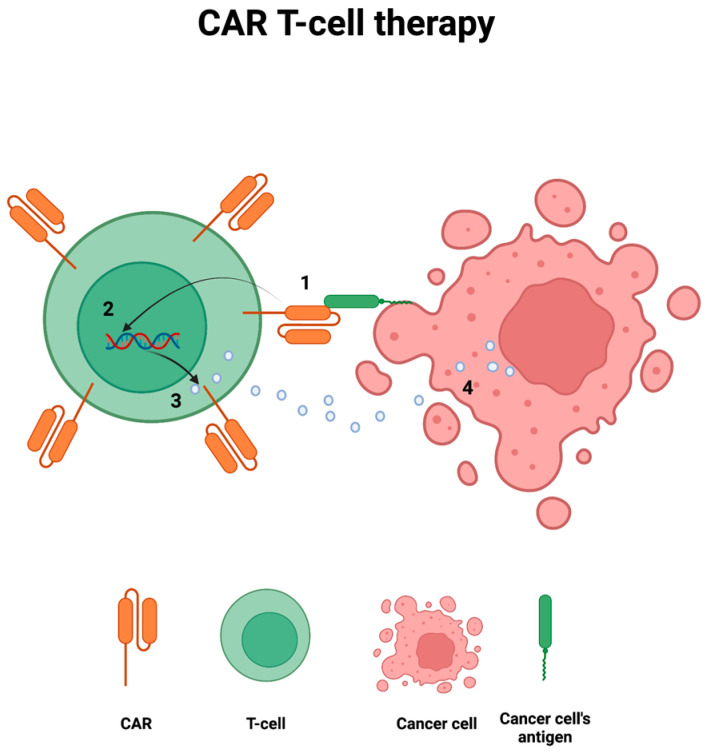
CAR T-cell therapies for cancer treatment. CAR T interacts with cancer cell’s antigen (1); it then activates a co-stimulatory signal which stimulates the synthesis, transcription, and translation of perforins and granzymes (2, 3) that ultimately kill the cancer cell (4).

**Figure 4 biomedicines-11-01406-f004:**
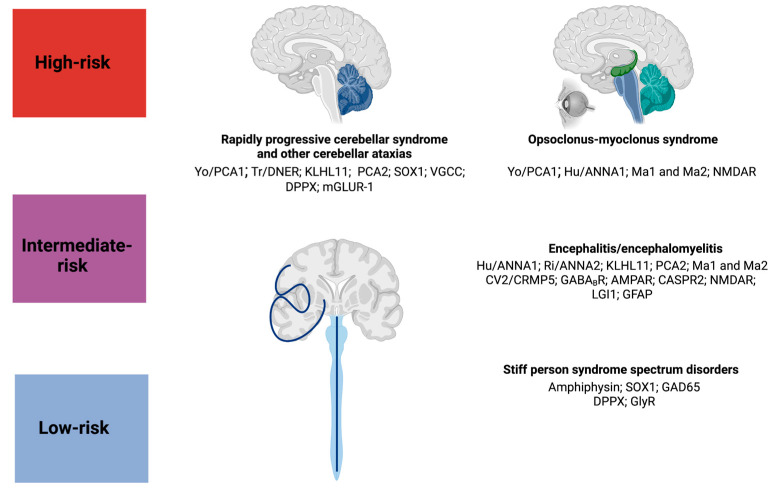
Main paraneoplastic syndromes and related antibodies. The figure illustrates the main paraneoplastic syndromes, their associated antibodies, and their risk of associated malignancy.

**Table 1 biomedicines-11-01406-t001:** Main antibodies and their associated paraneoplastic neurological syndromes.

Antibody	Cancer Type	Paraneoplastic Neurological Syndrome
**High-Risk**		
Anti-Yo/PCA1	Breast cancer, ovarian cancer	Rapidly progressive cerebellar syndrome, OMS
Anti-Hu/ANNA1	SCLC, NSCLC	Limbic encephalitis, encephalomyelitis, OMS
Anti-Ri/ANNA2	Breast cancer in women, lung cancer in men	Brainstem encephalitis
Anti-Tr/DNER	Hodgkin lymphoma	Rapidly progressive cerebellar syndrome
Anti- KLHL11	Testicular germ cell cancer	Brainstem encephalitis, rapidly progressive cerebellar syndrome
Anti-PCA2	SCLC, NSCLC, breast cancer	Rapidly progressive cerebellar syndrome, encephalitis
Anti-Ma1 and Ma2	Testicular cancer and NSCLC	Limbic and brainstem encephalitis, OMS
Anti-CV2/CRMP5	SCLC and thymoma	Encephalitis, encephalomyelitis
Anti-Amphiphysin	SCLC and breast cancer	SPSD
Anti-SOX1	SCLC	Rapidly progressive cerebellar syndrome, SPSD
**Intermediate-risk**		
Anti-GABA_B_R	SCLC	Limbic encephalitis
Anti-AMPAR	SCLC and thymoma	Limbic encephalitis
Anti-CASPR2	Thymoma	Morvan syndrome, Limbic encephalitis
Anti-NMDAR	Ovarian or extra-ovarian teratoma	Encephalitis, OMS
Anti-VGCC	SCLC	Rapidly progressive cerebellar syndrome
**Low-risk**		
Anti-LGI1	Thymoma	Limbic encephalitis
Anti-GAD65	SCLC and thymoma (rare)	SPSD
Anti-DPPX	Lymphoma	SPSD, PERM, cerebellar ataxia
Anti-GFAP	Ovarian teratoma and adenocarcinoma	Meningoencephalitis
Anti-GlyR	Lymphoma, thymoma, and lung cancer	SPSD, PERM
Anti-mGLUR-1	Lymphoma	Cerebellar ataxia

NSCLC, non-small-cell lung cancer; OMS, opsoclonus–myoclonus syndrome; PERM, progressive encephalomyelitis with rigidity and myoclonus; SCLC, small-cell lung cancer; SPSD, Stiff-person spectrum disorders.

## Data Availability

Not applicable.

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
