# Peer review of "Paraneoplastic Neurological Syndromes of the Central Nervous System: Pathophysiology, Diagnosis, and Treatment"

_biomedicines, 2023, doi:10.3390/biomedicines11051406_

Round 1

Reviewer 1 Report

The present review article by Marsili and colleagues, entitled ‘Paraneoplastic neurological syndromes of the central nervous system’ is a well-written and useful summary on the status of knowledge of diagnostic features and possible treatment of paraneoplastic neurological syndromes (PNS) of the central nervous system (CNS), based on clinical, neuroimaging and antibody testing. For this purpose, here Authors discussed the role of specific antibodies in confirming a PNS and guiding the search for a cancer type, and, finally, summarized first- and second-line immune-treatments as well as the symptomatic treatments to relief patients symptoms.

In general, I think the idea of this manuscript is really interesting and the authors’ fascinating observations on this timely topic may be of interest to the readers of Biomedicines. However, some comments, as well as some crucial evidence that should be included to support the author’s argumentation, needed to be addressed to improve the quality of the manuscript, its adequacy, and its readability prior to the publication in the present form, in particular reshaping parts of the Introduction and Methods sections by adding more evidence and theoretical constructs.

Please consider the following comments:

A graphical abstract that will visually summarize the main findings of the manuscript is highly recommended.

Abstract: According to the Journal’s guidelines, this section should be presented as a short summary of about 200 words maximum that objectively represents the article. It should let readers get the gist or essence of the manuscript quickly, prepare the readers to follow the detailed information, analyses, and arguments in the full paper and, most of all, it should help readers remember key points from your paper. Please, consider rewrite this paragraph following these instructions.

Keywords: I would suggest changing the keyword ‘paraneoplastic’ into ‘Paraneoplastic neurological syndromes (PNS)’, which seems to be more appropriate and correct. 

The objectives of this study are generally clear and to the point; however, I believe that there are some ambiguous points that require clarification or refining. I think that authors here need to be explicit regarding how they operationally investigated the current clinical features, neurological manifestations and treatment of PNS, since this is the key aim of this review.

I would ask the Authors to clarify the criteria they decided to use for studies’ collection in their review: they should specify the requirements used to decide whether a study met the inclusion/exclusion criteria of the review, describe whether they included a balanced coverage of all information that is actually available, whether they have included the most recent and relevant studies and enough material to show the development and limitations in this field of interest. Finally, I believe that they should briefly present results of all statistical syntheses conducted.

Introduction: The ‘Introduction’ section is well-written and nicely presented, with a good balance of descriptive text and information about clinical manifestations of paraneoplastic syndromes at different levels of neuraxis. Neverthless, I believe that more information about neural substrates neuropathophysiolgy of these disorders, in particular on how related brainstem dysfunctions and dysautonomia, will provide a better and more accurate background to this topic. For example, it would be very useful to further focus on the association between neural-specific antibodies with oncological association, that may be described with various paraneoplastic including sensory neuronopathy, gastroparesis, and limbic encephalitis and possible cognitive decline (https://doi.org/10.3390/biomedicines10123189; https://doi.org/10.1016/j.neubiorev.2023.105163). This information may help deepening information on how blunted cardiovascular reactivity may serve as an index of psychological task disengagement in the motivated performance.

Treatment: In my opinion, this section would benefit from a re-organization. It should include more evidence on the most common approaches used in the treatment of PNS. In this regard, I would suggest to also focus on describing general approaches to therapy, like removal of the antigen source by treatment of the underlying tumors, and suppression of the immune response, and prognostic.

In my opinion, I think the ‘Conclusions’ paragraph would benefit from some thoughtful as well as in-depth considerations by the authors, because as it stands, it is very descriptive but not enough theoretical as a discussion should be. Authors should make an effort, trying to explain the theoretical implication as well as the translational application of their research.

I would ask the authors to include a proper ‘Limitations and future directions’ section before the end of the manuscript, in which authors can describe in detail and report all the technical issues brought to the surface.

References: Authors should consider revising the bibliography, as there are several incorrect citations. Indeed, according to the Journal’s guidelines, they should provide the abbreviated journal name in italics, the year of publication in bold, the volume number in italics for all the references. Also, some of the references are out of date:  please cite references from the last 10 years, particularly references from the recent 5 years.

Finally, what is the take-away message from this review article? It ends rather abruptly with no summary, no suggested directions or immediate challenges to overcome, no call to action, no indications of things we should stop trying, and only brief mention of alternative perspectives. What do the authors want us to take away from this paper?

Overall, I suggest submitting your work to an English native speaker to help with some grammar mistakes that can be found in different sections of the manuscript.

I hope that, after these careful revisions, this paper can meet the Journal’s high standards for publication. 

I am available for a new round of revision of this paper. I declare no conflict of interest regarding this manuscript. 

Best regards,

Reviewer

Overall, I suggest submitting your work to an English native speaker to help with some grammar mistakes that can be found in different sections of the manuscript.

Author Response

Reviewer#1

The present review article by Marsili and colleagues, entitled ‘Paraneoplastic neurological syndromes of the central nervous system’ is a well-written and useful summary on the status of knowledge of diagnostic features and possible treatment of paraneoplastic neurological syndromes (PNS) of the central nervous system (CNS), based on clinical, neuroimaging and antibody testing. For this purpose, here Authors discussed the role of specific antibodies in confirming a PNS and guiding the search for a cancer type, and, finally, summarized first- and second-line immune-treatments as well as the symptomatic treatments to relief patients’ symptoms.

In general, I think the idea of this manuscript is really interesting and the authors’ fascinating observations on this timely topic may be of interest to the readers of Biomedicines. However, some comments, as well as some crucial evidence that should be included to support the author’s argumentation, needed to be addressed to improve the quality of the manuscript, its adequacy, and its readability prior to the publication in the present form, in particular reshaping parts of the Introduction and Methods sections by adding more evidence and theoretical constructs.

Please consider the following comments:

  • Comment: A graphical abstract that will visually summarize the main findings of the manuscript is highly recommended.

            Response: We thank the Reviewer for the general positive overview of our manuscript and for the specific suggestion of the graphical abstract. The graphical abstract has now been added.

  • Comment: Abstract: According to the Journal’s guidelines, this section should be presented as a short summary of about 200 words maximum that objectively represents the article. It should let readers get the gist or essence of the manuscript quickly, prepare the readers to follow the detailed information, analyses, and arguments in the full paper and, most of all, it should help readers remember key points from your paper. Please, consider rewrite this paragraph following these instructions.

Response: We thank the Reviewer for this comment. The abstract section has now been fully revised and re-written, accordingly (Lines: 16-31). The word count is now 202.

  • Comment: Keywords: I would suggest changing the keyword ‘paraneoplastic’ into ‘Paraneoplastic neurological syndromes (PNS)’, which seems to be more appropriate and correct.

            Response: Following the Reviewer’s comment, we have now updated the keyword section and re-spelled the word ‘Paraneoplastic neurological syndromes (PNS).

  • Comment: The objectives of this study are generally clear and to the point; however, I believe that there are some ambiguous points that require clarification or refining. I think that authors here need to be explicit regarding how they operationally investigated the current clinical features, neurological manifestations, and treatment of PNS, since this is the key aim of this review.

Response: We have now better clarified the objectives of the present study as well as our main clinical criteria adopted in the narrative review of the available literature on the topic. We have added this piece of information in the introduction section, which has been thoroughly re-written (Lines: 112-130).

  • Comment: I would ask the Authors to clarify the criteria they decided to use for studies’ collection in their review: they should specify the requirements used to decide whether a study met the inclusion/exclusion criteria of the review, describe whether they included a balanced coverage of all information that is actually available, whether they have included the most recent and relevant studies and enough material to show the development and limitations in this field of interest. Finally, I believe that they should briefly present results of all statistical syntheses conducted.

            Response: We thank the Reviewer for pointing out this important aspect. We have now added a full new “Methods” section (Lines 144-165) highlighting the methodology applied in the present study. Finally, in the introduction and methods sections, we have better outlined that ours is a narrative review, as all the other manuscript belonging to the ongoing special issue “Immune-mediated neurological disorders.”

  • Comment: Introduction: The ‘Introduction’ section is well-written and nicely presented, with a good balance of descriptive text and information about clinical manifestations of paraneoplastic syndromes at different levels of neuraxis. Nevertheless, I believe that more information about neural substrates neuropathophysiolgy of these disorders, in particular on how related brainstem dysfunctions and dysautonomia, will provide a better and more accurate background to this topic. For example, it would be very useful to further focus on the association between neural-specific antibodies with oncological association, that may be described with various paraneoplastic including sensory neuronopathy, gastroparesis, and limbic encephalitis and possible cognitive decline (https://doi.org/10.3390/biomedicines10123189; https://doi.org/10.1016/j.neubiorev.2023.105163). This information may help deepening information on how blunted cardiovascular reactivity may serve as an index of psychological task disengagement in the motivated performance.

            Response: We thank the Reviewer for the insightful comment. We have now extensively re-written the introduction section and added more information on the pathophysiology of PNS, highlighting the important role of brainstem dysfunctions and dysautonomia. We also focused on the association between neural-specific antibodies with oncological association in multiple PNS. We have finally cited the manuscripts suggested by the Reviewer (Lines: 36-130).

  • Comment: Treatment: In my opinion, this section would benefit from a re-organization. It should include more evidence on the most common approaches used in the treatment of PNS. In this regard, I would suggest to also focus on describing general approaches to therapy, like removal of the antigen source by treatment of the underlying tumors, and suppression of the immune response, and prognostic.

Response: We have now expanded and reorganized the “Treatment” section, and added the prognostic aspects too, as required by the Reviewer (Lines: 605-702).

  • Comment: In my opinion, I think the ‘Conclusions’ paragraph would benefit from some thoughtful as well as in-depth considerations by the authors, because as it stands, it is very descriptive but not enough theoretical as a discussion should be. Authors should make an effort, trying to explain the theoretical implication as well as the translational application of their research.

            Response: We thank the Reviewer for this comment, and according also to the Editor’s recommendations, we have now added a whole new paragraph on the “Discussion” and then modified the “Conclusion and next steps” section (Lines: 703-764).

  • Comment: I would ask the authors to include a proper ‘Limitations and future directions’ section before the end of the manuscript, in which authors can describe in detail and report all the technical issues brought to the surface.

            Response: As suggested by the Reviewer, we have now added the “Limitations” section (Lines: 749-754) at the end of the “Discussion” section, and then modified the “Conclusion and next steps” section, as required (Lines: 755-764).

  • Comment: References: Authors should consider revising the bibliography, as there are several incorrect citations. Indeed, according to the Journal’s guidelines, they should provide the abbreviated journal name in italics, the year of publication in bold, the volume number in italics for all the references. Also, some of the references are out of date:  please cite references from the last 10 years, particularly references from the recent 5 years.

            Response: We have now thoroughly revised the bibliography (and updated it reaching a total number of citations of 150, as also required by the Editor), the appropriateness of the citations, and their style, as required. Also, we have prioritized the most recent publications and left the oldest ones (e.g., more than 10-year-old), only if strictly relevant (e.g., published in high-impact journals or if fundamental for the discussed topic) within the context of the debated topic.

  • Comment: Finally, what is the take-away message from this review article? It ends rather abruptly with no summary, no suggested directions, or immediate challenges to overcome, no call to action, no indications of things we should stop trying, and only brief mention of alternative perspectives. What do the authors want us to take away from this paper?

            Response: We have now added the take-home message of the present review in the conclusion section (Lines: 761-764).

  • Comment: Overall, I suggest submitting your work to an English native speaker to help with some grammar mistakes that can be found in different sections of the manuscript.

            Response: We have now thoroughly revised the English language as required, through the expertise of a native English speaker.

I hope that, after these careful revisions, this paper can meet the Journal’s high standards for publication. 

I am available for a new round of revision of this paper. I declare no conflict of interest regarding this manuscript. 

Best regards,

Reviewer

We hope the changes made satisfy the  Reviewer' s concerns and made this paper suitable for publication in Biomedicines.

Reviewer 2 Report

This is well-written and concise review of CNS paraneoplastic syndromes.  I have only a few minor recommendations for clarity.

Table 1 is probably not in the intent publishable format, but it needs to be formatted in such a way that the individual categories are clearly separated.

Line 150:  The word “evenience” is not a word in common English parlance.  I would recommend changing “this evenience” to “these clinical features”

Line 158: change “includes” to “include”

Line 163: change “best described” to “more commonly seen” ?  It’s not clear what” best described” means here.

Line 178: omit the first “associated” in “associated OMS-associated”

Lines 183-186.  The first sentence seems to contradict the second.  It may be OK to eliminate the first sentence altogether or combine it with the second.  In the first sentence it sounds as though brainstem encephalitis is followed by more generalized involvement, but in the second it says it may occur in isolation.

Lines 209-211:  How do changes in the corticospinal tracts on MRI “correspond” with amyotrophy and fasciculations, which you acknowledge as lower motor neurons findings?

Lines 330-331: “Antibodies associated with PERM, namely glycine receptor (GlyR) and DPPX antibodies, yield a lower risk for an underlying malignancy”.  Change this to “Patients with antibodies associated with PERM, namely glycine receptor (GlyR) and DPPX antibodies, have a lower risk for an underlying malignancy.”   The antibodies don’t yield a lower risk, the patients with those antibodies have lower risk.

Line 337: “autoimmune” should be changed to “idiopathic” or “idiopathic autoimmune”, since PNS are also autoimmune in nature, we just have a better idea of the trigger.

Line 342: change “multiple” to “numerous”

Line 348: change ”IrAES” to “IrAEs”

Lines 349-350: change “dermatological” to “integumentary”, and “rheumatological” to “musculo-skeletal”.

Lines 354-355: change “require excluding” to “and all alternative etiologies (e.g., carcinomatous meningitis) must be excluded”.

Line 375: change “may be severe and sometimes fatal” to “may have severe or fatal clinical courses”.

Line 395: “autoimmune” (see line 337) to “non-neoplastic-induced autoimmune”

Line 408: wouldn’t a score of 10 be the “greatest probability” rather than “a higher probability” particularly considering a score of 8 or above is considered as “definite PNS”?

Line 437:  change “only” to “and only”

Lines 438-439: change “the clinical presentation of PNS” to “a clinical presentation mimicking PNS”.

Line 440: change “and the markers of inflammation of the CNS” to “as well as the presence of evidence of CNS inflammation”.

Line 449: change “In PNS is” to “In PNS it is”

Line 459: change “by” to “of”’

Line 460: change “comes” to “come”

Line 461: change “single centers” to “single-center”

Line 463: change “for” to “with”

Line 470: omit the word “the” before “second-line”

Line 494: omit the first hyphen in ICI-s-induced

Line 496: change “treatment” to “treatments” or “therapies”

Minor changes are noted in the review.

Author Response

Reviewer#2

This is well-written and concise review of CNS paraneoplastic syndromes.  I have only a few minor recommendations for clarity.

Comment: Table 1 is probably not in the intent publishable format, but it needs to be formatted in such a way that the individual categories are clearly separated.

            Response: We thank the Reviewer for the positive and encouraging comments on our review article. We have now edited the Table 1 as required by the Reviewer.

Comment: Line 150:  The word “evenience” is not a word in common English parlance.  I would recommend changing “this evenience” to “these clinical features.”

            Response: We have now changed the wording of the sentence, as required by the Reviewer (Line 234).

Comment: Line 158: change “includes” to “include”

Response: We have changed the word spelling, as required (Line: 242).

Comment: Line 163: change “best described” to “more commonly seen”?  It’s not clear what” best described” means here.

Response: We agree with the Reviewer that the former wording was not clear, and we have rephrased the sentence, as suggested (Line 247).

Comment: Line 178: omit the first “associated” in “associated OMS-associated”

Response: We thank the Reviewer; we have now removed the first “associated” word (Line 262).

Comment: Lines 183-186.  The first sentence seems to contradict the second.  It may be OK to eliminate the first sentence altogether or combine it with the second.  In the first sentence it sounds as though brainstem encephalitis is followed by more generalized involvement, but in the second it says it may occur in isolation.

Response: We thank the Reviewer for pointing out this important aspect. We have now rephrased the sentence, as follows: “Brainstem encephalitis is characterized by prominent brainstem involvement, followed or not by multisystem neurologic dysfunction (e.g., in association with more widespread encephalitis or rapidly progressive cerebellar degeneration, as discussed above in Section 1)] [15].” (Lines: 273-276).

Comment: Lines 209-211:  How do changes in the corticospinal tracts on MRI “correspond” with amyotrophy and fasciculations, which you acknowledge as lower motor neurons findings?

Response: We have now changed the sentence as follows: “More recently, cases of anti-Ma2 encephalitis have been described with a motor syndrome characterized by proximal muscle weakness, head drop, and bulbar symptoms. Rarely, patients may develop atrophy or fasciculations in the upper extremities mimicking motor neuron disease. T2/FLAIR hyperintensities may be seen on brain MRI in the corticospinal tract [54]” (Lines: 298-302).

Comment: Lines 330-331: “Antibodies associated with PERM, namely glycine receptor (GlyR) and DPPX antibodies, yield a lower risk for an underlying malignancy”.  Change this to “Patients with antibodies associated with PERM, namely glycine receptor (GlyR) and DPPX antibodies, have a lower risk for an underlying malignancy.”   The antibodies don’t yield a lower risk, the patients with those antibodies have lower risk.

Response: We thank the Reviewer for this comment. We have now changed the sentence as suggested (Lines: 448-449).

Comment: Line 337: “autoimmune” should be changed to “idiopathic” or “idiopathic autoimmune”, since PNS are also autoimmune in nature, we just have a better idea of the trigger.

Response: We agree with the Reviewer, and we have now changed it to: “idiopathic,” as suggested. (Line 466).

Comment: Line 342: change “multiple” to “numerous”

Response: We have now changed the word, as required (Line: 472).

Comment: Line 348: change” IrAES” to “IrAEs”

Response: Thank you, we have corrected the spelling (Line: 478)

Comment: Lines 349-350: change “dermatological” to “integumentary”, and “rheumatological” to “musculo-skeletal”.

Response: Changed as required (Line 482-483).

Comment: Lines 354-355: change “require excluding” to “and all alternative etiologies (e.g., carcinomatous meningitis) must be excluded”.

Response: Changed as suggested (Lines: 484-485)

Comment: Line 375: change “may be severe and sometimes fatal” to “may have severe or fatal clinical courses”.

Response: Change done (Line: 506).

Comment: Line 395: “autoimmune” (see line 337) to “non-neoplastic-induced autoimmune”

Response: Changed as requested (Line 539)

Comment: Line 408: wouldn’t a score of 10 be the “greatest probability” rather than “a higher probability” particularly considering a score of 8 or above is considered as “definite PNS”?

Response: We agree with the Reviewer and have now rephrased as required (Line: 555).

Comment: Line 437:  change “only” to “and only”

Response: We thank the Reviewer for this comment. According to the Editor’s suggestions, we have modified the whole sentence to add a new reference (Hence, when in doubt, clinicians should always tested CSF. Antibodies to LGI-1 may be an exception, as they are often absent or present only in low-titers in CSF, and more frequently isolated in serum [119].) (Lines: 592-594).

Comment: Lines 438-439: change “the clinical presentation of PNS” to “a clinical presentation mimicking PNS”.

Response: Change addressed, as required (Line 595).

Comment: Line 440: change “and the markers of inflammation of the CNS” to “as well as the presence of evidence of CNS inflammation”.

Response: Changed as required (Line: 597).

Comment: Line 449: change “In PNS is” to “In PNS it is”

Response: Thank you, change addressed (Line 606).

Comment: Line 459: change “by” to “of”’

            Response: Changed the whole period, to improve English language (Lines: 623-626).

Comment: Line 460: change “comes” to “come”

Response: Done (Line 627).

Comment: Line 461: change “single centers” to “single-center”

Response: Done (Line 627).

Comment: Line 463: change “for” to “with”

Response: Changed the whole sentence to improve English language (Lines: 626-629).

Comment: Line 470: omit the word “the” before “second-line”

Response: Done (Line 691).

Comment: Line 494: omit the first hyphen in ICI-s-induced

Response: Done (Line 758).

Comment: Line 496: change “treatment” to “treatments” or “therapies”

Response: Changed to “treatments” (Line 760).

We hope the changes made satisfy the Reviewer' s concerns and made this paper suitable for publication in Biomedicines.

Round 2

Reviewer 1 Report

The authors did an excellent job clarifying all the questions I have raised in my previous round of review. Currently, this paper entitled ‘Paraneoplastic neurological syndromes of the central nervous system’ is a well-written, timely piece of research that described diagnostic features and possible treatment of paraneoplastic neurological syndromes (PNS) of the central nervous system (CNS), based on clinical, neuroimaging and antibody testing.

Overall, this is a timely and needed work. It is well researched and nicely written, therefore I believe that this paper does not need a further revision, therefore the manuscript meets the Journal’s high standards for publication.

I am always available for other reviews of such interesting and important articles.

Thank You for your work, Reviewer

Author Response

Thank you so much for your kind comments. We really appreciate your suggestions and believe that the manuscript has now significantly improved.